# The Role of Inflammation in Age-Associated Changes in Red Blood System

**DOI:** 10.3390/ijms24108944

**Published:** 2023-05-18

**Authors:** Eryk Wacka, Edyta Wawrzyniak-Gramacka, Anna Tylutka, Barbara Morawin, Marzena Gutowicz, Agnieszka Zembron-Lacny

**Affiliations:** Department of Applied and Clinical Physiology, Collegium Medicum University of Zielona Gora, 65-417 Zielona Gora, Poland; e.wacka@cm.uz.zgora.pl (E.W.); e.gramacka@cm.uz.zgora.pl (E.W.-G.); a.tylutka@cm.uz.zgora.pl (A.T.); b.morawin@cm.uz.zgora.pl (B.M.); m.gutowicz@cm.uz.zgora.pl (M.G.)

**Keywords:** anemia, cytokines, erythropoietin, hematopoietic progenitor cells, hepcidin, iron

## Abstract

Aging-related anemia contributes to frailty syndrome, cognitive decline and early mortality. The study aim was to evaluate inflammaging in relation to anemia as a prognostic indicator in affected older patients. The participants (73.0 ± 7.2 years) were allocated into anemic (*n* = 47) and non-anemic (*n* = 66) groups. The hematological variables RBC, MCV, MCH, RDW, iron and ferritin were significantly lower, whereas erythropoietin EPO and transferrin Tf tended toward higher values in the anemic group. Approx. 26% of individuals demonstrated transferrin saturation TfS < 20%, which clearly indicates age-related iron deficiency. The cut-off values for pro-inflammatory cytokine IL-1β, TNFα and hepcidin were 5.3 ng/mL, 97.7 ng/mL and 9.4 ng/mL, respectively. High IL-1β negatively affected Hb concentration (r_s_ = −0.581, *p* < 0.0001). Relatively high odds ratios were observed for IL-1β (OR = 72.374, 95%Cl 19.688–354.366) and peripheral blood mononuclear cells CD34 (OR = 3.264, 95%Cl 1.263–8.747) and CD38 (OR = 4.398, 95%Cl 1.701–11.906), which together indicates a higher probability of developing anemia. The results endorse the interplay between inflammatory status and iron metabolism and demonstrated a high usefulness of IL-1β in identification of the underlying causes of anemia, while CD34 and CD38 appeared useful in compensatory response assessment and, in the longer term, as part of a comprehensive approach to anemia monitoring in older adults.

## 1. Introduction

Anemia is a common hematological disorder that affects 17% of persons aged >65 years. The World Health Organization (WHO) defines anemia as a decreased number of erythrocytes and/or decreased hemoglobin (Hb) levels of <12.0 g/dL in women and <13.0 g/dL in men, and the definition dates back to the 1970s. Analyses of the American databases of National Health and Nutrition Examination Survey III [1] and the Scripps–Kaiser database [2] have suggested higher reference values for Hb to define anemia [3]. The optimal Hb levels of ≥13.7 g/dL for men and ≥12.6 g/dL for women have been described in connection with better survival in the Cardiovascular Health Study [4]. A population study by Culleton et al. demonstrated the levels of 13.0 to 15.0 g/dL for women and 14.0 to 17.0 g/dL for men as the optimal Hb values to avoid hospitalization and mortality in old age [5]. Wouters et al. [6] suggested that the definition of anemia in women over 60 years of age should include a modified value of Hb, i.e., <13.0 g/dL, which makes the definition comparable to the one applied to men. Nevertheless, the WHO definition remains the standard for the classification of anemia in older adults [7]. Normal Hb distribution varies not only with regard to age and sex, but it also depends on ethnicity and physiological status [3,8]. Penninx et al. [9] and Katsumi et al. [10] indicated the need to also identify a risk of physical decline in older adults by detection of borderline anemia, defined as a decreased Hb levels within 1 g/dL above WHO criteria. Additionally, it is noteworthy that anemia has been demonstrated to have a significant impact on survival in individuals older than 60 years of age [6].

In older adults, anemia can be divided into nutritional deficiency anemia, bleeding anemia, and unexplained anemia that might be caused by the reduced erythropoietin (EPO) activity, the progressive erythropoietin resistance of bone marrow erythroid progenitors and the chronic subclinical pro-inflammatory state [11]. Overall, one-third of older patients with anemia have a nutritional deficiency which mainly includes iron, folate or vitamin B12 deficiency, one-third have a chronic subclinical pro-inflammatory state and a chronic kidney disease, and one-third suffer from anemia of an unknown cause [12]. Understanding the pathophysiology of anemia in this population is crucial because it contributes to frailty syndrome and falls, cognitive decline, depression, functional ability deterioration and early mortality [10]. A prospective cohort analysis of 3758 patients aged 65 years and older showed that a new-onset anemia was associated with an increased mortality risk with a drop in Hb of 1 g/dL [13].

The pro-inflammatory state in older age, called inflammaging, is manifested by the release of a large number of inflammatory mediators that are produced to repair damage at the tissue level, such as: interleukins IL-1, IL-2, IL-6, IL-8, IL-12, IL-13, IL-15, IL-18, IL-22, IL-23, tumor necrosis factor α (TNFα) interferon-γ (IFN-γ) as pro-inflammatory cytokines, and IL-1Ra, IL-4, IL-10; transforming growth factor (TGF-β1) as anti-inflammatory cytokines; and also lipoxin A4 and heat shock proteins as cytokine mediators [14,15]. Inflammaging is a result of changes in the immune system also known as immunosenescence [16,17]. According to Minciullo et al. [17], inflammaging is a key to our understanding of the aging process, and anti-inflammaging may be one of the secrets of longevity. Thus, it is important to intervene more quickly and multidimensionally in anemia of inflammation.

The inflammatory molecules produce adverse effects on the cells of the hematological system, and these include iron deficiency, reduced EPO production and elevated phagocytosis of erythrocytes by hepatic and splenic macrophages, and also enhanced eryptosis by oxidative stress in the circulation [10,18]. First, systemic inflammation induces the production of hepcidin, the major regulator of iron homeostasis, in hepatocytes through interleukin 6 (IL-6) production. Hepcidin both reduces intestinal iron absorption and releases iron from the macrophages. The recycling of iron from senescent erythrocytes by macrophages accounts for more than 90% of the daily iron requirements for erythropoiesis. Older adults with anemia of inflammation have higher hepcidin levels than their counterparts without anemia [19,20]. The second pathogenic factor in anemia of inflammation is the reduced EPO level and progressive EPO resistance of bone marrow erythroid progenitors. Tumor necrosis factor α and interleukin 1β (IL-1β) inhibit the production of EPO through GATA-binding protein 2 and nuclear factor kappa B transcription [21]. In addition, IL-1β inhibits the binding of hepatocyte nuclear factor 4 to the enhancer of the EPO gene, whereas TNFα stimulates the secretion of microRNA122 from the liver, which reduces EPO gene expression in the kidney [22,23]. Interferon-γ reduces the expression of the EPO receptor in erythroid progenitor cells, inhibits their differentiation and shortens the lifespan of erythrocytes [24]. Thus, TNFα, IL-1β, IL-6 and IFN-γ seem to be responsible for impaired EPO synthesis in the development of anemia associated with inflammation (Figure 1).

Anemia contributes to age-associated diseases and significantly affects the health-related quality of life [6]. In older individuals, even mild anemia is associated with an increased risk of falls, decreased physical performance, longer and more frequent stays in hospital and increased mortality [5,25,26,27]. Despite its clinical importance, evidence-based guidelines for anemia management in older adults are yet to be developed [28]. The underlying cause of anemia still remains unexplained in 30% of cases [29]. It is important to identify the pathways that control age-related inflammation across multiple systems in order to understand and prevent anemia [14]. Therefore, the present study focused primarily on the evaluation of the major feature of inflammation in relation to anemia, thereby indicating therapeutic strategies to counteract the pathophysiological effect of age-associated changes in red blood cell system.

## 2. Results

### 2.1. Participants and Body Composition

In anemic group, the participants were significantly older than in the non-anemic group (Table 1). Moreover, the number of males was higher in the anemic group (*n* = 33) than in non-anemic group (*n* = 18). BMI ranged from 20.7 to 40.8 kg/m^2^. For age, AUC was relatively high in the tested model, anemic vs. non-anemic (AUC = 0.722, sensitivity 62.1%, specificity 74.2%), which indicates a good prognostic value of age to distinguish the individuals with anemia from no-anemia ones. The odds ratio (OR) of >1 means that individuals aged >71.5 years (cut-off value) demonstrated a four-fold higher probability of developing anemia (OR = 4.712, 95%Cl 1.961–11.959). Approximately 55% of the study participants had a normal body mass (18.5–24.9 kg/m^2^), 29.2% were classified as overweight (25.0–29.9 kg/m^2^) and 15.9% as obese (≥30 kg/m^2^) [30].

### 2.2. Hematological Variables

White blood cells, neutrophils, lymphocytes and monocytes were within the referential ranges, and they did not differ significantly between the anemic and non-anemic groups (Table 2). According to the Culleton et al. classification [5], 42% of our study individuals were described as anemic (Hb < 13.0 g/dL for women and Hb < 14.0 g/dL for men). Admittedly, men with a low concentration of Hb constituted 70% of the anemic group, but no significant differences were observed between women (13.36 ± 0.84 g/dL) and men (13.37 ± 1.65 g/dL). In the total observed population, men demonstrated significantly lower levels of iron and transferrin (Tf) (84.38 ± 26.72 µg/dL and 2.75 ± 1.16 mg/dL) than women (100.82 ± 31.71 µg/dL and 3.17 ± 1.99 mg/dL), whereas ferritin levels did not differ between sexes. Hb was inversely correlated with IL-1β (r_s_ = −0.581, *p* < 0.0001), which confirms the impairment of Hb synthesis under inflammation [19,20,22,23]. The other hematological variables such as RBC, MCV, MCH, RDV, iron and ferritin were significantly lower, whereas EPO and Tf were within the referential ranges but tended toward higher values in the anemic group. Approximately 26% of the study participants demonstrated transferrin saturation (TfS) < 20%, and 20% had ferritin <10–15 ng/mL, which together indicates age-related iron deficiency. For TfS, AUC is relatively high in the tested model, anemic vs. non-anemic group (AUC = 0.756, sensitivity 62.2%, specificity 83.0%, OR = 7.76, 95%Cl 2.64–25.23), which indicates a good affiliation of TfS to the observed population compared to other markers of iron deficiency.

### 2.3. Biochemical Variables

The total cholesterol and lipoproteins have been proven to be the strongest biomarkers of aging [31]. High levels of TG > 150 mg/dL were found in 23% of the study participants, whereas high levels of TC > 200 mg/dL, LDL > 130 mg/dL and non-HDL > 130 mg/dL were recorded in 63% of them. HDL lipoproteins were significantly lower in the anemic group than in the non-anemic group (Table 3). As is the case with cholesterol, an elevated glucose level is a known biomarker of aging, and it is associated with alterations in metabolic and hormonal functioning, including altered expression of cellular insulin receptors and glucose transporter units in target tissues [31]. Some participants (*n* = 9) demonstrated an increased glucose level >115 mg/dL without being diagnosed with diabetes. Serum bilirubin level, as a biomarker of red blood cells aging or disintegration, was recorded within normal ranges, but it was two-fold higher in the anemic group, which may enhance erythropoiesis expressed as a high EPO concentration in that group (Table 4).

### 2.4. Inflammatory Variables

Most observational studies and clinical trials have used high-sensitivity CRP as a biochemical marker of inflammation because it is relatively stable and easy to measure. In our study, CRP concentration was recorded at ≥3 mg/L in 43% of the patients, which indicates the subclinical inflammatory state according to the reference values for the older adults described by Wyczalkowska-Tomasik et al. [32]. NPT concentration reached high values >10 nmol/L in the majority of our patients, indicating a crucial role of NPT in inflammaging. However, CRP and NPT levels did not differ between anemic and non-anemic groups in a statistically significant manner (Table 4). Classical cytokine biomarkers of aging including IL-1β, IL-6 and TNFα were higher in the anemic group. The most considerable changes concerned IL-1β, TNFα and HPC, which were approx. 1.5–3-fold higher in the anemic group than in the non-anemic group (Table 4). The diagnostic ability of hematological and inflammatory variables was determined by receiver operating characteristic (ROC) curve analysis. The results of the ROC and OR analysis of age and inflammatory variables, i.e., IL-1β, TNFα and HPC, indicate a potential diagnostic value for clinical prognosis for patients with age-associated anemia. The optimal threshold values corresponded to 71.5 years for age, 5.3 ng/mL for IL-1β, 97.7 ng/mL for TNFα and 9.4 ng/mL for HPC (Table 5). The highest average odds ratio was observed for IL-1β (AUC = 0.523, OR = 72.374, 95%Cl 19.688–354.366), which indicates this cytokine as a good marker of inflammation-related anemia. The highest sensitivity (87.9%) and specificity (91.5%) were also observed for IL-1β, which, in turn, indicates a low level of false-positive results during diagnostic procedure. It was surprising that IL-6, called “a cytokine for gerontologists” [17], showed poor diagnostic usefulness to distinguish the individuals with anemia from no-anemia ones (AUC = 0.523, sensitivity 62.2%, specificity 51.1%, OR = 1.939, 95%Cl 0.848–4.489) compared to other cytokines (Table 5). Interestingly, HPC level was more than twice as high in the study for men (16.79 ± 16.08 ng/mL) compared to women (7.00 ± 6.99 ng/mL), which resulted in low iron concentrations in men. For HPC, AUC was relatively high in the tested model, anemic vs. non-anemic (AUC = 0.735, sensitivity 84.8%, specificity 59.6%, OR = 8.069, 95%Cl 3.131–22.397), which indicates a good affiliation of this pathogenic factor to the observed population.

### 2.5. Peripheral Blood Mononuclear Cells

The levels of peripheral blood mononuclear cells CD34 and CD38 were significantly higher in the anemic group (23.40 ± 14.63 ng/mL and 1.73 ± 0.83 ng/mL) than the non-anemic group (17.81 ± 10.57 ng/mL and 1.37 ± 0.74 ng/mL), which proves that these cells participate in the compensation mechanism in age-associated anemia. The levels of CD34 and CD38 changed simultaneously (Figure 2). For CD34 and CD38, AUC was moderately high in the tested model, anemic vs. non-anemic (AUC = 0.600, sensitivity 76.8%, specificity 50.0% for CD34, and AUC = 0.657, sensitivity 76.8%, specificity 57.8% for CD38), which indicates their moderate diagnostic utility in patients with anemia. However, the OR values for CD34 (OR = 3.264, 95%Cl 1.263–8.747) and CD38 (OR = 4.398, 95%Cl 1.701–11.906) suggested a significantly higher probability of developing anemia in patients with cut-off values >19.1 ng/mL for CD34 and >1.4 ng/mL for CD38.

## 3. Discussion

In this study, low Hb concentration was observed to be associated with subclinical, chronic inflammation, exhibited by high levels of IL-1β and TNFα, and their inverse correlations (Hb/IL-1β r_s_ = −0.581, *p* < 0.0001; Hb/TNFα r_s_ = −0.225, *p* < 0.01) in the analyzed population. Inflammaging is typically manifested by the release of a large number of pro-inflammatory mediators which are produced to repair damage at the tissue level; however, their persistently elevated levels contribute to the damaging catabolic effects on aging tissues, leading to degradation and damage acceleration and eventually causing a decline in mental and physical performance [17,33]. In the large InCHIANTI study of more than 1300 individuals, the levels of IL-6, IL-1ra, IL-18, CRP and fibrinogen were all elevated in patients aged over 65; yet, when adjusted for cardiovascular risk factors and morbidity, the elevation was reduced [34]. The unexplained anemia cohort (36% of all the anemic population) was found to have higher levels of pro-inflammatory markers and higher resistance of bone marrow erythroid progenitors to erythropoietin compared to non-anemic controls [8,35]. In our study population, hematological variables such as RBC, MCV, MCH, RDV, iron and ferritin were significantly lower in the group with low Hb concentration, and 26% of individuals demonstrated TfS < 20%, which together indicates age-related anemia. EPO tended toward high values in anemic group, and it was two-fold higher in men (13.06 ± 12.65 mU/mL) than women (7.96 ± 4.63 mU/mL). A sex breakdown was reported in the majority of studies concerning anemia in older adults [36].

Population studies reported the mean and median Hb concentrations to be higher by 12% in age-matched men in comparison to women [37]. Altered end-organ response to circulating testosterone or EPO, reduced erythropoietic reserve or decreased marrow reactivity may account for the observed increase in anemia with age, particularly in older men. Sex differences, however, may reflect the prevalence of undetected chronic disease or other underlying causes of anemia in these populations [38]. The difference in Hb levels observed between our study women (13.36 ± 0.84 g/dL) and men (13.37 ± 1.65 g/dL) was not significant in the total population, which confirms the validity of the suggestion put forward by Wouters et al. [6] that the value of Hb < 13.0 g/dL should be applied for both sexes aged over 60 years.

The studies from the nineties demonstrated that EPO blood level in anemic patients suffering from inflammation was often decreased in relation to Hb concentration. To date, various proinflammatory cytokines have been tested for their action on the EPO gene expression in isolated perfused rat kidneys and in human hepatoma cell cultures, and it is mainly IL-1β and TNFα that have been indicated as molecules responsible for the defect in EPO production in severe systemic and renal inflammatory diseases [35]. However, in studies conducted in older adults, the elevated EPO production can potentially be triggered by an impaired oxygen delivery to the kidney, which can result from various pathophysiologic mechanisms such as anemia, hypoperfusion caused by renal arteriosclerosis, decreased renal blood flow or heart failure, decreased oxygen saturation due to diseases such as chronic obstructive pulmonary disease, and eryptosis-related decrease in blood oxygen capacity [18,39]. Generally, there are two relevant hypotheses, i.e., a decreased production of EPO or an insufficient bone marrow response to EPO, but neither of them has been satisfactorily resolved by experimental evidence [18]. The elevated EPO level recorded in the anemic group (12.77 ± 4.32 mU/mL) when compared to the non-anemic group (8.05 ± 3.04 mU/mL) clearly emphasizes the need for further studies to investigate whether the elevated EPO levels observed in anemic patients are due to EPO resistance. One potential study design could involve comparing the EPO levels of anemic patients to those of younger patients with uncomplicated iron deficiency and the same degree of anemia. This would facilitate a direct comparison of EPO response between two groups with similar anemia severity but different age and disease status. If the EPO response is blunted in anemic patients compared to younger patients with iron deficiency, this would suggest the presence of EPO resistance in the anemic group. Such studies could provide valuable insights into the underlying mechanisms of anemia and EPO resistance, which could ultimately inform the development of more effective management strategies.

The pro-inflammatory state is thought to elicit a chronic elevation of circulating hepcidin leading to impaired availability of plasma iron, limiting Hb synthesis and eventually causing anemia of inflammation. Whether HPC plays a role in anemia in older adults from the general population has been debated [40]. HPC binds to the membrane protein ferroportin, an iron efflux channel on the surface of absorptive enterocytes, macrophages and hepatocytes, and induces its internalization and degradation in lysosomes, thereby sequestering iron in the cytoplasm of these cells [41]. As a result, serum iron level decreases rapidly [40]. The anemic group in our study demonstrated higher levels of HPC than their counterparts without anemia. Earlier, the Leiden 85-plus Study showed elevated serum hepcidin levels, unlike the InCHIANTI study, which did not demonstrate an increase in urinary hepcidin levels in older adults with anemia of inflammation [34,40]. In turn, the Val Borbera study [42] and the Nijmegen Biomedical Studies [43] showed that hepcidin levels in pre-menopausal women were nearly 50% lower than in men of corresponding ages. After the menopause, hepcidin levels tended to be similar in both sexes. Interestingly, in our study, HPC levels remained at lower levels in women (7.00 ± 6.99 ng/mL) than in men (16.79 ± 16.08 ng/mL). The cut-off value for HPC ≥ 9.4 ng/mL significantly increased the risk of anemia in the observed population. The results of the ROC analysis of HPC, as was the case in cytokines IL-1β and TNFα, ranged between 0.6 and 1.0, indicating a potential diagnostic value for clinical prognosis for patients with age-associated anemia.

In older populations, changes in ferritin concentration do not always correlate with variations in iron stores because ferritin is an acute phase protein and is affected by inflammatory processes irrespective of the iron store status. Lower levels of ferritin were recorded in our anemic group (16.04 ± 21.82 ng/mL) when compared to the non-anemic group (38.27 ± 44.91 ng/mL), which proves the depletion of iron stores. However, diagnostic capability of ferritin varies across different studies with regard to cut-off points [44,45,46]. In our study, the cut-off value for ferritin was 16.4 ng/mL. This result indicates that in the individuals aged 60 years and older, iron deficiency anemia may develop with higher levels of serum ferritin than 10 ng/mL (reference value for women) and 15 ng/mL (reference value for men). Hence, the conventional serum ferritin cut-off value used to diagnose iron deficiency anemia in young adults is not applicable for the older adults. This creates difficulties in the diagnostic capability of the ferritin test in anemia diagnostics. Our observations warrant further investigations to determine the ferritin level of more reliable diagnostic properties. A systematic review of 55 studies found some variations in ferritin test results across populations with and without inflammatory processes, liver disease or neoplastic diseases [47]. For ferritin, AUC was moderate in the tested model anemic vs. non-anemic (AUC = 0.652, sensitivity 60.6%, specificity 74.5%), which indicates mediocre diagnostic utility, whereas for the TfS, AUC was relatively high (AUC = 0.756, sensitivity 62.2%, specificity 83.0%, OR = 7.76, 95%Cl 2.64–25.23), which indicates a good affiliation of TfS to the observed population. Studies have showed that TfS is less affected by inflammatory processes and may therefore be more accurate and reliable than serum ferritin, particularly in conditions with elevated inflammatory state [48].

The last decade of studies has indicated the hematopoietic stem cell’s ability to differentiate into various blood cell types, such as red and white blood cells and platelets. These stem cells can self-renew, and they can generate fully developed blood cells [49]. CD34 is expressed on the surface of peripheral blood mononuclear cells such as hematopoietic progenitor cells and endothelial cells [49,50], and studies on CD34 function suggested that it might play a role in cell adhesion and signal transduction [51]. CD38 is expressed on early and activated hematopoietic cells [50], and it is particularly associated with immune cells such as B cells, T cells and natural killer cells. While CD38 expression has been observed in hematopoietic stem cells, its role in these cells is still not fully understood. However, studies suggest that CD38 may play a role in the regulation of hematopoietic stem cells differentiation and proliferation [52]. CD34 and CD38 proteins are usually used as surface markers to identify hematopoietic stem cells and leukemic stem cells [53]. This study is the first to have shown high CD34 and CD38 expression in older adults with anemia of inflammation (Figure 2). Although the available literature on CD34 and CD38 involvement in anemia is limited, and the exact mechanisms underlying the observed differences in CD34 and CD38 levels are not fully understood, the engagement of several factors may already be identified. One possibility is that the higher expression of these markers in the anemic group reflected a compensatory response to the reduced oxygen-carrying capacity of the blood [54,55], indicated by odd ratios >1 for CD34 and CD38. Another possibility is that high levels of CD34 and CD38 might be a marker of increased inflammation or oxidative stress in the anemic group. Both anemia and oxidative stress are known to promote the release of pro-inflammatory cytokines and reactive oxygen species, which can activate immune cells and potentially induce the expression of CD34 and CD38 [56]. Therefore, further studies into the underlying mechanisms of CD34 and CD38 expression in inflammatory-related anemia are needed to fully understand their potential usefulness as indicators of anemia severity and associated pathologies.

## 4. Materials and Methods

### 4.1. Participants

One hundred and thirteen individuals were recruited from the University of the Third Age in Zielona Gora (Poland), which is an organization encouraging adults over 60 years of age to stay active by participating in many educational programs. The current health status of the participants was assessed on the basis of medical records at a routine follow-up visit to a primary care physician. The inclusion criteria included the age of 65 years or older and the same access to medical healthcare provided by the same medical center. On the basis of the medical interview, the following exclusion criteria were applied: acute infectious and autoimmunological diseases, uncontrolled hypertension and/or diabetes, oncologic diseases, neurodegenerative diseases and folate (<1.8 ng/mL) or vitamin B12 (<140 pg/mL) deficiency. The participants (females *n* = 62, males *n* = 51) aged 73.0 ± 7.2 years were included in the project, and they were allocated to two groups, i.e., anemic (*n* = 47) and non-anemic (*n* = 66), as a conventional marker of anemia, according to the reference values <13 g/dL for women and <14 g/dL for men, as defined by Culleton et al. [5]. The medications taken by the participants included antihypertensive (84%) and hypolipidemic (10%) drugs as well as anticoagulants including anti-platelet agents (15%). All the subjects were informed of the aim of the study and signed a written consent to participate in the project. The study protocol was approved by the Regional Bioethics Commissions (Regional Medical Chamber in Zielona Gora, No. 04/133/2020, University of Zielona Gora No. UZ/19/2021), in accordance with the Helsinki Declaration.

### 4.2. Blood Sampling

Fasting blood samples were collected from the median cubital vein in the morning between 8.00 and 10.00 using S-Monovette tubes (Sarstedt AG & Co. KG, Nümbrecht, Germany). The whole-blood samples were placed into specimen tubes containing EDTA and were immediately analyzed. For the other biochemical analyses, blood samples were centrifuged at 3000 rpm for 10 min, and aliquots of serum were stored at −80 °C. The average intra-assay coefficients of variation (intra-assay CV) for the used enzyme immunoassay tests (ELISA) were <8%. All samples were analyzed in duplicate in a single assay to avoid inter-assay variability.

### 4.3. Hematological Variables

Hematological parameters including total white blood cell count (WBC), red blood cell count (RBC), platelet count, differential white cell count (neutrophils, lymphocytes, monocytes, eosinophils and basophils) and hemoglobin concentration (Hb) were determined by Sysmex XN-1000 (Sysmex Europe Gmbh, Norderstedt, Germany). Iron and ferritin levels were determined using BM200 Biomaxima (Lublin, Poland). The levels of transferrin (Tf) and erythropoietin were identified by using ELISA kits from SunRed Biotechnology Company (Shanghai, China) with detection limit of 0.05 mg/mL and 0.526 mU/mL. The percent of transferrin saturation (TfS) was calculated as follows: TfS% = (serum iron level/transferrin level) × 100.

### 4.4. Biochemical Variables

Serum triglycerides (TG), total cholesterol (TC), high-density lipoproteins (HDL) and low-density lipoproteins (LDL) were determined using BM200 Biomaxima (Lublin, Poland). The non-HDL cholesterol was calculated by subtracting HDL from total cholesterol concentration. Oxidized low-density lipoprotein (oxLDL) was determined using ELISA kits from SunRed Biotechnology Company (Shanghai, China) with detection limit at 30.3 ng/mL. Glucose and bilirubin were determined using BM200 Biomaxima (Lublin, Poland).

### 4.5. Inflammatory Variables and Peripheral Blood Mononuclear Cells

C-reactive protein (CRP), neopterin (NPT) and hepcidin (HPC) concentrations were determined using highly sensitive ELISA kits from DRG International Inc. (Springfield Township, Cincinnati, OH, USA). The detection limits for CRP, NPT and HPC were set at 0.001 mg/L, 0.7 nmol/L and 0.153 ng/mL, respectively. IL-1β, IL-6 and TNFα concentrations were determined by using ELISE kits from SunRed Biotechnology Company (Shanghai, China) with detection limits of 0.028 ng/mL, 1.867 pg/mL and 2.782 ng/mL, respectively. The levels of peripheral blood mononuclear cells expressing CD34 and CD38 were determined by using ELISA kits from SunRed Biotechnology Company (Shanghai, China) with detection limit of 0.167 ng/mL and 0.009 ng/mL, respectively.

### 4.6. Statistical Analysis

Statistical analyses were performed using R 4.2.1 software [57]. Levene and the Shapiro–Wilk tests were used to evaluate the homogeneity of variances and the normality of the distributions, respectively. The significant differences in mean values between the groups (anemic and non-anemic) were assessed via Student’s *t*-test. If the homogeneity and normality assumptions were violated, the non-parametric Mann–Whitney test was used. The optimal clinical stratification thresholds (cut-off values) were based on receiver operating characteristic (ROC) curves by calculating the Youden index. The odds ratio (OR) was calculated using odds ratio function from the epitools package in R. Spearman’s rank correlation coefficient (r_s_) was used to investigate the relationships between hematological and inflammatory markers. The variables were reported as mean value ± standard deviation (SD) or median (Me). Statistical significance was set at *p* < 0.05.

## 5. Conclusions

The mechanisms underlying low Hb levels in older adults are multifactorial and complex. Our study suggested that the underlying mechanisms involve subclinical chronic low-grade inflammation, bone marrow resistance to EPO, and changes in hepcidin levels, ultimately affecting iron metabolism and resulting in lower serum iron levels. IL-1β, TNFα and hepcidin revealed the highest diagnostic value, whereas CD34 and CD38 showed usefulness in evaluation of the compensatory response to the reduced oxygen-carrying capacity in inflammatory-related anemia. A comprehensive understanding of these mechanisms is crucial for accurate diagnosis and effective management of anemia in older patients. Further research is warranted to fully elucidate the complex interplay between hematological and inflammatory factors. Our improved understanding of the pathophysiological mechanisms underlying anemia could serve as crucial intervention targets, leading to enhanced survival and overall quality of life in the older population.

## Figures and Tables

**Figure 1 ijms-24-08944-f001:**
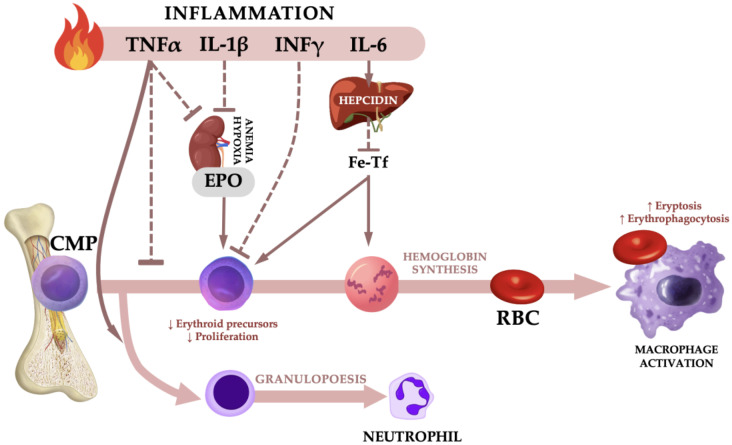
Mechanisms associated with anemia of inflammation: TNFα, tumor necrosis factor α; IL-1β, interleukin 1β; IFN-γ, interferon γ; IL-6, interleukin 6; EPO, erythropoietin; Fe-Tf, iron-transferrin; CMP, common myeloid progenitor; RBC, red blood cell. Dashed line represents inhibition. The upward arrow symbolizes growth or an upward trend, while the downward arrow signifies a reduction or a downward trend.

**Figure 2 ijms-24-08944-f002:**
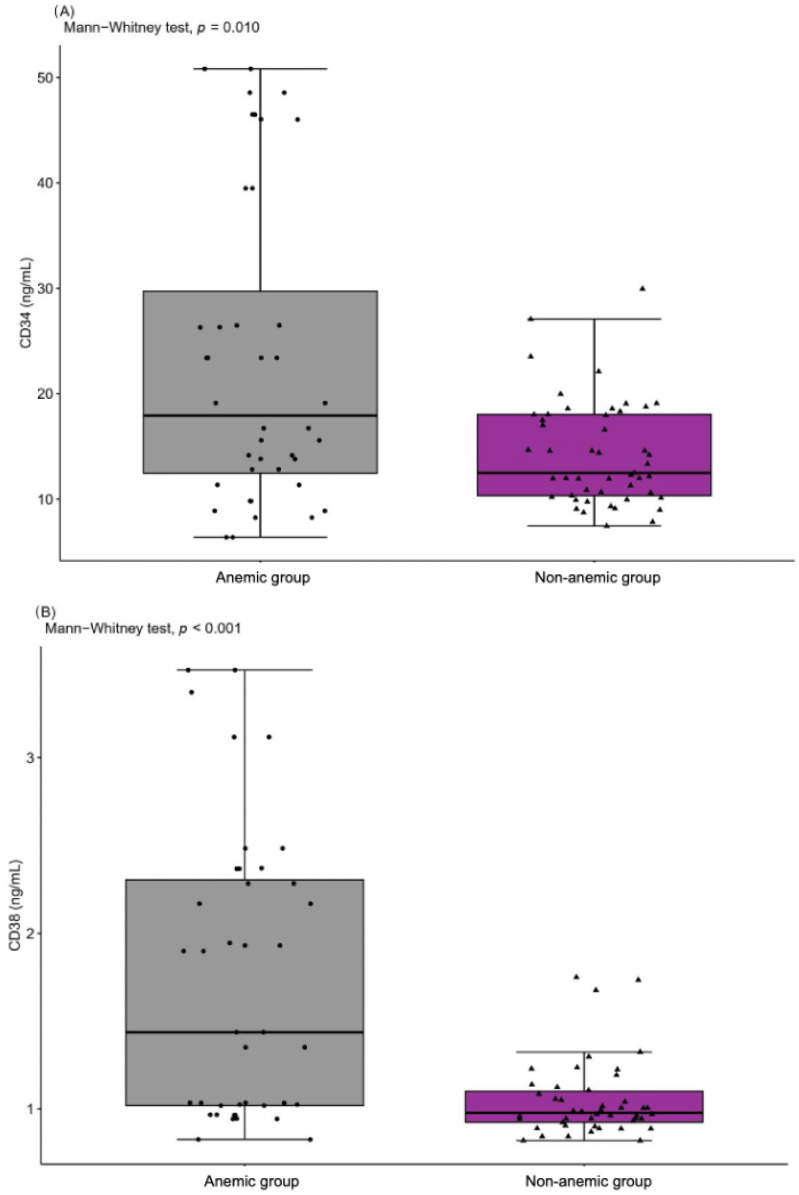
The levels of peripheral blood mononuclear cells CD34 (**A**) and CD38 (**B**) in anemic (represented by black circles) and non-anemic (represented by black triangles) groups.

**Table 1 ijms-24-08944-t001:** Age and anthropometric characteristics.

	Anemic *n* = 47	Non-Anemic *n* = 66	Anemic vs. Non-Anemic *p* Level
	Mean ± SD (Me)	Mean ± SD (Me)	
Age (years)	77.3 ± 5.0 (78.0)	71.8 ± 7.6 (71.0)	<0.001
Weight (kg)	74.7 ± 12.7 (73.0)	69.7 ± 10.5 (67.5)	<0.001
Height (cm)	167.3 ± 6.5 (168.0)	160.2 ± 6.0 (159.0)	<0.001
BMI (kg/m^2^)	26.6 ± 3.6 (26.1)	27.3 ± 3.7 (27.1)	0.765

Abbreviations: BMI, body mass index.

**Table 2 ijms-24-08944-t002:** Hematological variables.

	Reference Values	Anemic *n* = 47	Non-Anemic *n* = 66	Anemic vs. Non-Anemic *p* Level
		Mean ± SD (Me)	Mean ± SD (Me)	
WBC (10^3^/µL)	4.0–10.2	6.37 ± 2.12 (5.67)	6.25 ± 1.72 (6.08)	0.937
Neutrophils (10^3^/µL)	2.0–6.9	4.14 ± 1.87 (3.90)	3.59 ± 1.44 (3.29)	0.052
Lymphocytes (10^3^/µL)	0.6–3.4	1.80 ± 0.77 (1.66)	1.89 ± 0.63 (1.84)	0.207
Monocytes (10^3^/µL)	0.00–0.90	0.44 ± 0.19 (0.42)	0.56 ± 0.29 (0.51)	0.055
RBC (10^6^/µL)	F 4.0–5.5 M 4.5–6.6	4.20 ± 0.39 (4.24)	4.69 ± 0.53 (4.64)	<0.001
Hb (g/dL)	F 12.5–16.0 M 13.5–18.0	12.50 ± 0.98 (12.70)	14.26 ± 0.90 (14.05)	<0.001
Hct%	F 37–47 M 40.0–51.0	36.25 ± 3.56 (37.0)	42.58 ± 2.87 (42.25)	<0.001
MCV (fL)	82–92	86.42 ± 6.30 (88.00)	91.30 ± 4.99 (91.95)	<0.001
MCH (pg)	27–33	29.81 ± 1.56 (29.80)	30.58 ± 1.61 (30.75)	<0.001
MCHC (g/dL)	32–36	34.61 ± 1.93 (33.70)	33.52 ± 1.16 (33.35)	0.015
RDW%	11.5–14.5	14.56 ± 2.44 (14.30)	14.75 ± 0.98 (14.85)	<0.001
Platelets (10^3^/µL)	140–420	219 ± 56 (211)	229 ± 57 (226)	0.417
MPV (fL)	7.5–10.5	8.85 ± 1.77 (9.10)	7.26 ± 1.16 (7.11)	<0.001
Iron (µg/dL)	F 37–145 M 58–158	78.22 ± 24.76 (76.20)	104.21 ± 30.27 (101.60)	<0.001
Ferritin (ng/mL)	F 10–200 M 15–400	16.04 ± 21.82 (8.60)	38.27 ± 44.91 (23.43)	<0.001
Tf (mg/dL)	200–400	393 ± 184 (306)	302 ± 99 (298)	0.096
TfS%	20–50	23.40 ± 11.24 (21.14)	37.09 ± 15.90 (37.07)	<0.001
EPO (mU/mL)	4.3–29	12.77 ± 4.32 (11.02)	8.05 ± 3.04 (7.24)	0.197

Abbreviations: WBC, white blood cells; RBC, red blood cells; Hb, hemoglobin; Hct, hematocrit; MCV, mean cell volume; MCH, mean corpuscular hemoglobin; MCHC, mean corpuscular hemoglobin concentration; RDW, red cell distribution width; MPV, mean platelet volume; Tf, transferrin; TfS, transferrin saturation; EPO, erythropoietin.

**Table 3 ijms-24-08944-t003:** Lipoprotein-lipid profile and other biochemical variables.

	Reference Values	Anemic *n* = 47	Non-Anemic *n* = 66	Anemic vs. Non-Anemic *p* Level
		Mean ± SD (Me)	Mean ± SD (Me)	
TG (mg/dL)	<150	114.43 ± 36.69 (117.40)	143.78 ± 31.77 (139.23)	<0.001
TC (mg/dL)	<200	194.80 ± 48.87 (187.96)	228.90 ± 38.67 (228.70)	<0.001
LDL (mg/dL)	<130	91.37 ± 35.93 (90.58)	87.89 ± 30.84 (89.05)	0.864
oxLDL (mg/dL)	–	0.031 ± 0.034 (0.016)	0.038 ± 0.041 (0.014)	0.822
HDL (mg/dL)	desirable > 60	61.25 ± 13.74 (60.70)	71.48 ± 19.49 (69.59)	<0.001
non-HDL (mg/dL)	<130	133.56 ± 45.62 (125.16)	157.42 ± 43.37 (160.04)	<0.001
Glucose (mg/dL)	60–115	94.44 ± 28.17 (86.50)	93.33 ± 20.61 (93.40)	<0.001
Bilirubin (mg/dL)	<1.0	0.24 ± 0.18 (0.22)	0.12 ± 0.13 (0.07)	<0.001

Abbreviations: TG, triglycerides; TC, total cholesterol; LDL, low-density lipoproteins; oxLDL, oxidized low-density lipoprotein; HDL, high-density lipoproteins; non-HDL, cholesterol calculated by subtracting the HDL value from a TC.

**Table 4 ijms-24-08944-t004:** Inflammatory variables.

	Anemic *n* = 47	Non-Anemic *n* = 66	Anemic vs. Non-Anemic *p* Level
	Mean ± SD (Me)	Mean ± SD (Me)	
CRP (mg/L)	3.12 ± 2.34 (3.12)	2.92 ± 2.29 (2.54)	0.843
NPT (nmol/L)	23.16 ± 11.30 (22.88)	19.95 ± 5.86 (19.95)	0.205
HPC (ng/mL)	16.84 ± 13.88 (15.60)	6.42 ± 3.29 (6.08)	<0.001
IL-1β (ng/mL)	11.69 ± 9.83 (8.19)	2.55 ± 3.64 (1.21)	<0.001
IL-6 (pg/mL)	97.22 ± 68.28 (75.78)	74.98 ± 27.27 (74.72)	0.195
TNFα (ng/mL)	121.50 ± 69.82 (112.00)	85.65 ± 59.64 (82.41)	<0.001

Abbreviations: CRP, C-reactivity protein; NPT, neopterin; HPC, hepcidin; IL-1β, interleukin 1β; IL-6, interleukin 6; TNFα, tumor necrosis factor α.

**Table 5 ijms-24-08944-t005:** The statistical characteristics of the ROC curve for hematological and inflammatory variables and the odds ratio for indications determined by the cut-off values (*n* = 113).

Variables	AUC	Cut-off Value	Sensitivity (%)	Specificity (%)	OR	95%CI
Iron (µg/dL)	0.745	88.6	68.2	72.3	5.510	2.292–13.937
Ferritin (ng/mL)	0.652	16.4	60.6	74.5	4.424	1.845–11.193
Tf (mg/mL)	0.622	3.2	78.4	48.9	3.422	1.208–10.546
TfS (mg/dL)	0.756	32.5	62.2	83.0	7.76	2.64–25.23
EPO (mU/mL)	0.583	8.48	59.1	48.9	0.725	0.318–1.643
HPC (ng/mL)	0.735	9.4	84.8	59.6	8.069	3.131–22.397
IL-1β (ng/mL)	0.922	5.3	87.9	91.5	72.374	19.688–354.366
IL-6 (pg/mL)	0.523	75.5	65.2	51.1	1.939	0.848–4.489
TNFα (ng/mL)	0.685	97.7	74.2	66.0	5.488	2.292–13.741

Abbreviations: AUC, the area under the curve; cut-off value the optimal threshold value for clinical stratification; OR, odds ratio; 95%CI, confidence interval.

## Data Availability

The data used to support the findings of this study are available from the corresponding author upon request.

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
