# Peer review of "The Role of Inflammation in Age-Associated Changes in Red Blood System"

_ijms, 2023, doi:10.3390/ijms24108944_

Round 1
Reviewer 1 Report
TAuthors have suggested the underlying mechanisms involving subclinical 430 chronic low-grade inflammation, bone marrow resistance to EPO, and changes in hepcidin levels, ultimately affecting iron metabolism and resulting in lower serum ferrous 432 levels. This research could serve as 439 crucial intervention targets, leading to enhanced survival and overall quality of life. The abstract section is well-elaborated. The introduction section has a sufficient number of citations with good content. The topic is highly significant. The methodology is well described with proper reference. All the graphs are clear and the data presented are statistically significant. However, there are some concerns:
The diagram quality is not clear.
The results need to be rewritten with references.
Please check the grammatical errors in the manuscript.
The conclusion section is very nicely explained deciphering the significance of the study in a very concise manner. Altogether, this manuscript can be accepted after minor revisions.
Please check the grammatical errors in the manuscript.
Reviewer 2 Report
This is an interesting study of anemia in older individuals and its correlations. Although the population is of moderate size, it is very well defined. A number of aspects of the study require expanded analysis/interpretation.
1. The term "inflammageing" needs to be defined when it first appears.
2. See comments below about the use of the term "ferrous".
3. Rather than saying "seventies of the previous century" the authors should just say "1970s".
4. The authors should note that the WHO criteria for anemia were not designed to identify clinically relevant anemia in specific (particularly European/North American/more developed) countries but rather to represent an agreed-upon minimum below which they would be no argument patient was anemic.
5. Rather than referring to the two study groups as "LowHb" "HighHb", they need to refer to them as "anemic" or "not anemic".
6. Table 1 addressing body composition issues has a number of potential interpretation problems. The authors raise the reasonable interpretation that excess visceral fat may contribute to anemia, for which there are some data in the literature. However, these data need to be reanalyzed controlling for patient sex. Men are disproportionately represented in the anemia population, and some of these factors may reflect that. For example, it appears that greater height is a predictor of anemia. Is that an artifact of sex distribution between the populations?
7. Interpretation of serum ferritin levels requires more discussion/nuance. The ferritin range stated by the authors would include a significant proportion of iron deficient individuals. While ferritin can indeed be falsely elevated out of proportion to iron stores by inflammation, a low ferritin is always an indicator of iron deficiency. While the normal ranges stated by the authors may reflect the population distribution used by the laboratory, studies correlating serum ferritin with bone marrow iron stores (the gold standard for iron deficiency) clearly demonstrate that absence of reticuloendothelial iron stores occurs when the ferritin is less than 25-30 ng/mL, whether in men or in women. In this light, it would appear that a significant portion of both groups are iron deficient, although of course it will be worse in the anemic group.
8. More detailed discussion of EPO levels is required. EPO levels tended to be higher in the anemic group. This is actually the physiologic expectation. In the presence of anemia, the EPO-producing cells in the kidney are less well oxygenated and increase production of hypoxia-inducing factor that in turn drives EPO production. So this finding does not of itself demonstrate EPO resistance, although it could reflect EPO resistance. What would demonstrate EPO resistance would be comparing the EPO levels of anemic patients to those of uncomplicated younger patients with iron deficiency and the same degree of anemia, and seeing if the EPO response is blunted. This is the characteristic finding of anemia in inflammation – an EPO level which is increased but not as increased as would be predicted in anemia without inflammation. Obviously that particular analysis is probably beyond the scope of the current study, but the concept should be cited.
9. It is not clear that the lipid/biochemical variable tables adds anything useful.
10. The CD34/CD38 data are interesting and potentially useful. Technically what is being measured is not progenitor cells but rather peripheral blood mononuclear cells expressing CD34 or CD38. Some of the cells in this population will actually be specific progenitors.
The term “ferrous” is used inappropriately. Ferrous refers to the oxidative state of iron. The term they actually want to use is “iron”.
Reviewer 3 Report
The paper presented for evaluation concerns the assessment of the influence of age-related inflammation on hemoglobin levels. The "introduction" part introduces the reader to the discussed issues. In my opinion, an abstract is an excess of numerical data. I would suggest the authors to approach the presented correlations more carefully. Despite the statistical significance, the correlation coefficients are weak, as can be seen from the distribution of points in the graphs that lack confidence intervals. Legends are missing in box-whisker charts. However, the novelty of the work raises the most objections, and some of the results seem to be obvious.
Round 2
Reviewer 2 Report
The authors have responded appropriately to reviewer comments.